# Ecological Responses of *Nannophya koreana* (Odonata: Libellulidae) to Temperature: Following Converse Bergmann’s Rule

**DOI:** 10.3390/biology11060830

**Published:** 2022-05-27

**Authors:** Cha Young Lee, Min Kyung Kim, Dong-Gun Kim

**Affiliations:** 1Institute of Environmental Ecology, Sahmyook University, Seoul 01795, Korea; ishursain@gmail.com (C.Y.L.); min4190@naver.com (M.K.K.); 2Department of Bio & Environment Technology, Seoul Women’s University, Seoul 01797, Korea; 3Smith College of Liberal Arts, Sahmyook University, Seoul 01795, Korea

**Keywords:** body size, conservation, dragonfly, growth rate, local adaptation, temperature–size rule

## Abstract

**Simple Summary:**

Bergmann’s rule explains the phenomenon where populations and species of larger sizes are found at higher latitudes and colder environments, whereas populations and species of smaller size are found at lower latitudes and in warmer regions. In insects, adult sizes tend to be smaller in warmer environments than at cooler temperatures and higher latitudes; the response is called the temperature–size rule. *Nannophya koreana* is an endangered species in Korea and represents a flagship species for wetland conservation. We found that the body size of the larvae was smaller in a cold-water-temperature region than in a warm-water-temperature area, which is contrary to the rules mentioned above. The two regions were geographically close to each other, with no differences in air temperature and precipitation. We identified the reasons for the difference in water temperature between the two regions and established the relationship between temperature and body size in *N. koreana*. In addition, we analyzed how *N. koreana* compensated for low water temperature to maintain its life cycle, which is known as univoltine.

**Abstract:**

Ecological rules such as Bergmann’s rule and the temperature–size rule state that body-size decline is a universal response to warm temperatures in both homeotherms and poikilotherms. In the present study, we investigated the biological responses of *Nannophya koreana*, an endangered dragonfly species in Korea, by comparing body size in two habitats with large differences in water temperature, Mungyong-si (MG, terraced paddy fields) and Muui-do (MU, a mountainous wetland). To conserve the dragonfly populations, the collected larvae were photographed and released, and their head widths and body lengths were measured. There was no difference in the annual mean air temperature and precipitation between the two sites; however, the annual mean water temperature was substantially lower in MU than in MG. There was little difference in larval head width between the two sites; however, body length in the MU population was smaller than that in the MG population. Larval growth rate per 100-degree-days was 0.75 mm for MG and 1.16 for MU. The relationship between temperature and body size of *N. koreana* larvae showed opposite trends to Bergmann’s rule and the temperature–size rule. Since the larval growth period during a year in MU was shorter than that in MG, the MU population potentially exhibits a higher growth rate as a mechanism of compensating for the low water temperature. Our study established the relationship between temperature and body size of *N. koreana* in two wetlands that had an obvious difference in water temperature despite being geographically close. The results highlight the importance of considering detailed factors such as habitat type when studying the temperature–size responses of organisms.

## 1. Introduction

Body size of an organism is a fundamental trait that influences life history, fitness, behavior, resource use, and predator–prey relationships [1,2,3,4]. Differences in body size are affected by biotic and abiotic factors and vary across geographical regions [5,6,7]. One of the most classic ecogeographical patterns of body size is Bergmann’s rule, which states that body size increases with a decrease in temperature or an increase in latitude in homeotherms [8,9]. Similarly, the temperature–size rule (TSR) is universally known in poikilotherms, where the developmental rate is faster at lower latitudes and warmer temperatures; therefore, individuals developing in warm conditions will be smaller than those developing in cool conditions, yielding a pattern that conforms to Bergmann’s rule [2,4]. In insects, in response to shorter development times at higher latitudes, species with a univoltine life cycle can shift to a semivoltine life cycle. As a result, greater body size is likely to be observed in semivoltine populations due to their longer development period [7,10].

Various studies have been conducted on TSR in relation to climate change in many species [11,12]. Recently, however, studies showing a positive relationship between body size and temperature, with smaller body sizes in cooler climates, have been reported in poikilotherms; the relationship is referred to as the converse Bergmann’s rule [6,13,14,15,16]. Some investigations have revealed no geographical body size patterns [17,18,19,20,21,22]. Therefore, it is now clear that the body size response to warming temperatures does not have an absolute pattern. Such variability could be attributed to variability in voltinism, season length, habitat type, and food quality [2]. At higher latitudes, univoltine species have relatively small body sizes, to compensate for rapid development and growth rates with decreasing season length. At the point where two years are required to complete one generation, the body size increases instead of opting for compensatory growth [10,23]. Such latitude–body size patterns are also used to evaluate the possible strength of a compensatory mechanism [7].

Insect body sizes vary along ecological gradients, and a larger body size can buffer against harsh environments, such as starvation or desiccation [24]. However, being large is not always advantageous; for example, maintaining a large body size requires more resources and may impose a cost of predation or parasitism [25]. In odonates, their large size increases their vulnerability to predators and requires that the wetland retains water throughout the larval period [26,27].

Biological traits of organisms appear to be a result of developmental, morphological, physiological, and behavioral adaptations of organisms to their environment [28]. Aquatic invertebrates are highly affected by several parameters, including water temperature, and are likely to undergo various ecological changes, such as changes in their phenology, ontology, and fitness, as a result of environmental differences. The types of habitats in wetlands vary depending on the surrounding environment, and the distribution, physiology, and morphology of aquatic insects inhabiting wetlands are influenced by the habitat type [29]. In particular, dragonflies (Insecta: Odonata) are considered flagship taxa in wetlands and are indicator species for wetland conservation and health evaluation [30]. In Korea, habitat and population density studies have been conducted for many years to conserve the endangered dragonfly species, *Nannophya koreana* (Odonata: Libellulidae) [31,32,33,34]. Recently, a survey was conducted to collect ecological information on *N. koreana* in Muui-do, one of the habitats of *N. koreana* and a protected area for forest genetic resource conservation. The *N. koreana* population in Muui-do has exhibited ecological patterns that are distinct from those reported in previous research [33,35].

*N. koreana* belongs to the genus *Nannophya*, which includes the world’s smallest dragonflies, with a body length of approximately 15 mm [31,33,35,36]. *N. koreana* was first recorded in Korea in 1957 and has been recognized as a synonym of *N. pygmaea* Rambur, which is widely distributed throughout northern Australia and tropical and subtropical areas of Southeast Asia, India, China, Japan, and Korea [31,33,34]. However, a phylogenetic study using molecular characteristics (mitochondrial DNA, cytochrome oxidase c subunit I [COI]) determined that *N. koreana* and *N. pygmaea* are distinct species; therefore, the species inhabiting Korea and Japan turned out to be *N. koreana* [32]. *N. koreana* prefers a habitat with a low water depth of 2.5–10 cm, and the habitats are decreasing because of terrestrialization by climate change and habitat destruction by human activities [34]. Therefore, the species is designated as an endangered species, category level II, by the Ministry of Environment of Korea in 1988 [36]. Numerous studies have been conducted to study its distribution, habitat, life history, and habitat conservation and restoration in Korea [31,32,33,34,35,36,37,38,39].

Initially, it was suggested that the Muui-do population was a novel subspecies of *N. koreana,* but all *N. koreana*, including the Muui-do population, has been confirmed to belong to the same species [40]. Therefore, in the present study, we hypothesize that the short growing season in Muui-do, where the habitat temperature is low, imposes constraints on *N. koreana* body size development. Our aims were to establish the intraspecific temperature–size relationship of *N. koreana* and analyze the ecological responses of dragonflies to low water temperature. We explore whether *N. koreana* conforms to Bergmann’s rule and TSR by investigating the relationship between habitat temperature and larval body size. In addition, we analyze the ecological response of *N. koreana* to different habitat environments and the strategies and mechanisms it adopts to maintain its life cycle and compensate for low water temperature habitat conditions.

## 2. Materials and Methods

### 2.1. Study Sites

Although there are more than 20 known habitats of *N. koreana* in Korea, fewer than five have been confirmed to exist until recently [34]. Two were selected for a comparative study of the ecological responses of *N. koreana* (Figure 1). The first area was located in Mungyong-si, Gyeongsangbuk-do (MG) (36° 33 15.9′ N, 128° 00 20.2′ E, alt. 243 m a.s.l.). MG is a terraced paddy field (0.8 ha) that has been left unused for three to five years. Water depth was maintained at 5–10 cm. The substrate was mainly composed of silt and sand, with abundant organic detritus. The wetland vegetation included *Juncus effusus* and *Persicaria thunbergii.* The other study site was located 170 km northwest of MG and is a small wetland located at Horyonggoksan Mountain in Muui-do, Jung-gu, Incheon (MU) (37° 22 57.2′ N, 126° 24 53.4′ E, alt. 28 m a.s.l.). The wetland (0.1 ha) and the surrounding 1.5 ha area have been designated as a protected region by the national government because of its high ecological conservation value with rare and endangered species. The wetland source is spring water, and the water depth is maintained below 10 cm. The substrate was mainly composed of fine sand and silt. Rare plant species, such as *Iris ensata* var. *spontanea*, *Utricularia bifida*, and *Utricularia racemose*, inhabit the area.

### 2.2. Temperature Measurement

The water temperature was recorded at 2 h intervals for MG and 1 h intervals for MU, using a water temperature data logger (UTBI-001, Onset Computer Corporation, Bourne, MA, USA) during the investigation period. A data logger was installed on the bottom surface of the wetland and care was taken to avoid direct sunlight. The air temperature data were obtained from a weather station of the Korean Meteorological Administration (KMA, https://data.kma.go.kr (accessed on 8 January 2022)). The distances to the nearest weather station were 3.7 km for MG and 1.3 km for MU, respectively.

Water and air temperatures were converted to degree-days (DD) using the rectangle method [33,41], as shown in Equation (1):Rectangle DD = (T_max_ + T_min_)/2 − T_b_(1)
where DD is degree-days, T_max_ and T_min_ are the daily maximum temperature and minimum temperature, respectively, and T_b_ is the base temperature for egg development. The T_b_ value was 14.3 °C, as derived from previous studies [33,38]. Relationships between water temperature, as recorded by data loggers at the sites, and data from the weather stations were estimated by linear regression models using IBM SPSS 25.0 (IBM Corp., Armonk, NY, USA). These regressions were used to extrapolate the values of missing temperature data during the study period at each site.

### 2.3. Sampling and Size Measurement

Investigations of *N. koreana* in MG were conducted every 1–2 months from June 2006 to July 2007. Sampling was conducted every two weeks during the emergence period in 2007. In MU, investigations were conducted once a month from June 2019 to June 2021, except during winter. At each study area, *N. koreana* adults were counted with the naked eye using the line transect (25 × 4 m) method. Larvae were qualitatively investigated using a hand net sampler (stainless steel, 40 × 20 cm; mesh size 0.25 mm, Kitchen-Art Co., Ltd., Incheon, Korea). Because *N. koreana* is an endangered species designated by the Ministry of Environment of Korea, close-up images of the collected larvae were captured using a digital camera (PowerShot SX70HS, Canon, Tokyo, Japan), and the larvae were released again to protect the population. A ruler was photographed with the larvae for use as a reference unit when measuring larvae size with an image processing program. Larval head width (HW) was measured in dorsal view to obtain maximum width, usually across the compound eyes. Body length (BL) was measured from the anterior head to the terminal abdomen, including anal appendages (Figure 2). HW and BL were estimated using ImageJ v1.33 (U.S. National Institutes of Health, Bethesda, MD, USA).

### 2.4. Data Analysis

The environmental conditions at the two study sites, including air temperature, water temperature, and monthly precipitation, were compared using a paired *t*-test during the overlapping study period (August to July of the following year). Larval size data were analyzed using the Hcap program in MATLAB R2015a (The MathWorks, Natick, CA, USA) developed by Logan et al. [42]. Hcap is a useful method for determining the larval instar separation [43,44,45,46,47]. In the present study, the number of larval groups was determined instead of the larval instar. Subsequently, the method proposed by Gaines et al. [48] based on Dyar’s rule [49] was used to analyze the fit between the group number and the natural log of the mean HW measurements. Dyar’s ratios were then derived for all groups by taking the mean from one group and dividing it by the mean of the previous group.

To compare the larval size at each study site, we calculated the BL/HW ratio of larvae and performed a *t*-test on the two populations using this value. In addition, the HW and BL of the two populations in each group were compared using *t*-tests.

Odonate larvae cope with winter by seasonal regulation and diapause, and the larval growth and development rates are influenced by photoperiod and temperature [50]. In addition, odonates have variable voltinism according to the temperature [10]. Therefore, we attempted to analyze the variability of voltinism and compensatory growth by comparing the larval body size and growth rate of the two regional populations based on temperature and body size data. The accumulated degree-days (ADD) required for the development of *N. koreana* were calculated from 1 January of the year. In the present study, we used 14.3 °C as the T_b_ [35,38]. The growth rate of larvae was calculated using BL and ADD, as suggested in a previous study [35]:(2)$$\mathrm{Growth}\;\mathrm{rate}=\left[\overset{n}{\underset{i=1}{\sum }}\left\{\left(BL_{i+1}-BL_{i}\right)/\left(ADD_{i+1}-ADD_{i}\right)\right\}\right]/k-1$$
where $BL_{i}$ is the mean BL of the population collected at the ith investigation, $ADD_{i}$ is the ADD up to the ith investigation, and k is the number of investigations used in growth rate calculation. The results of the investigation during adult flying periods and the winter season were excluded from the growth rate calculation because in the adult flying periods, the larvae of the previous generation and the newborn larvae for the next generation were mixed, causing confusion in the estimation of the growth rate. During the winter season, the degree days did not accumulate because the water temperature was lower than the T_b_ value.

To estimate the ADD required for the flight peak of *N. koreana* adults, daily DD was calculated by fixing the T_b_ value and using a cumulative Weibull function [51] as suggested by Pruess [52]:(3)$$f\left(x\right)=1-\exp\left[-{\left(x⁄a\right)}^{b}\right]$$
where f(x) is the cumulative frequency distribution of the number of adults at x as the ADD, and parameters a and b represent the scale and shape of the function, respectively. Parameters a and b were calculated using SigmaPlot v14.0 (Systat Software, Inc., Richmond, CA, USA) using a sigmoidal curve-fitting tool with Weibull 2 parameters. The fitted functions based on the data from the two study sites were compared using the two-sample Kolmogorov–Smirnov test. The median date of adult flight (50% of the adults having flight in the flight period) was assumed to be the oviposition date. The *N. koreana eggs* needed 150 DD to hatch [35,38]; therefore, the date that satisfied 150 DD from the oviposition date was estimated as the hatching date. All statistical analyses were performed using IBM SPSS Statistics 25.0 (IBM Corp., Armonk, NY, USA).

## 3. Results

### 3.1. Environmental Conditions

Annual mean air temperatures during the overlapping study period (MG: from August 2006 to July 2007; and MU: from August 2019 to July 2020) were 12.67 ± 0.49 °C (mean ± standard error [SE]) and 12.66 ± 0.46 °C in MG and MU, respectively, and there was no significant difference between the two sites (paired *t*-test, *t* = 0.039, *p* = 0.969, *df* = 364; Figure 3a). Annual mean water temperatures during the same period were 21.07 ± 0.49 °C and 12.60 ± 0.45 °C in MG and MU, respectively. The water temperature in MG was significantly higher than that in MU (paired *t*-test, *t* = 29.627, *p* < 0.001, *df* = 364; Figure 3a). Monthly precipitations in MG and MU were 77.83 ± 23.01 mm and 87.33 ± 26.47 mm, respectively, and there was no significant difference between the two sites (paired *t*-test, *t* = −0.467, *p* = 0.650, *df* = 11; Figure 3b).

In MG, the aggregate set of water temperatures recorded by the data logger between 2006 and 2007 was significantly correlated with air temperature data from regional weather stations (*r*^2^ = 0.94, *p* < 0.001; Figure 4a). The daily mean water temperature in MU was estimated in the same manner (*r*^2^ = 0.90, *p* < 0.001; Figure 4b).

### 3.2. Grouping and Larval Size Distribution

A total of 186 and 144 larvae were sampled during each study period in MG and MU, respectively. The HW ranged from 1.18 to 3.21 mm and from 1.05 to 3.15 at MG and MU, respectively. Because the HW ranges of the two study sites were similar, Hcap analysis was performed to distinguish the larval groups by pooling the HW data of the two sites. As a result of the analysis, the frequency distribution of HW showed four distinct peaks, representing four larval groups (Figure 5a). The number of individuals, mean, and SE for each group are presented in Table 1. Larval HW ranged from 1.05 to 3.21 mm and difference in mean HW among the groups was significant (misclassification probabilities *p* < 0.05; Table 1). The Dyar’s ratio ranged from 1.22 to 1.35 (Table 1). The logarithm of HW plotted against the number of groups resulted in perfect geometric larval growth in each group (*r*^2^ = 0.99, *p* < 0.01; Figure 5b).

The BL ranged from 2.76 to 10.69 mm and from 3.18 to 9.29 at MG and MU, respectively, and a scatter plot for HW and BL of the two populations showed different distributions (Figure 6). The body sizes of the two populations were compared by calculating the BL to HW ratio, and the BL of the MG population was greater than that of the MU population (*t*-test, *t* = 16.499, *p* < 0.001, *df* = 317.129). The HW and BL of the two populations were compared for each larval group using the *t*-test. In group I, there were no statistically significant differences in HW (*t* = −0.738, *p* = 0.449, *df* = 46) or BL (*t* = 1.675, *p* = 0.101, *df* = 46) between the two populations. In groups II and III, there were no differences in HW (group II: *t* = −1.439, *p* = 0.155, *df* = 59; group III: *t* = −0.095, *p* = 0.925, *df* = 88), whereas the BL of the MG population was greater than that of the MU population (group II: *t* = 7.278, *p* < 0.001, *df* = 59; group III: *t* = 7.040, *p* < 0.001, *df* = 88) (Table 2). In Group IV, the HW of the MG population was less than that of the MU population (*t* = −4.760, *p* < 0.001, *df* = 129), but the BL of the MG population was greater than that of the MU population (*t* = 9.276, *p* < 0.001, *df* = 93.687) (Table 2).

### 3.3. Life History

At MG, the greatest number of larvae, 30 individuals, were collected on 17 June 2006, and the fewest, three individuals, were collected on 7 July 2006. The highest proportion of group I among the collected larvae was observed in October 2006, and the highest proportion of group IV was observed in May 2007. Adults appeared from the end of May to the beginning of August, the density was highest in June, and many were found until early August. In the case of MU, the most larvae, 23 individuals, were collected on 17 June 2021, and the fewest larvae, one individual, were collected on 13 August 2019. In August 2020, only group I larvae were collected from October to the spring of the following year, and the ratio of larvae collected from October to spring of the following year was high in group IV. The flight period of adults was from June to early August (Figure 7), and the density was the highest in June and rarely found after August.

At both study sites, group I newly appeared in June, but was not observed in May, indicating that the larvae of the previous generation and descendants were mixed in June–July. In MG, *N. koreana* larvae overwintered in various body sizes, whereas larvae in MU tended to grow their body sizes during the fall, prior to winter (Figure 7).

Growth rates of the larvae were estimated by calculating the mean BL of the population at each investigation and the difference in ADD between the periods of investigations, excluding the period when adults coexisted with the offspring larvae and the winter season. At MG, the mean BLs of the larvae collected in October and December 2006 were 5.54 and 6.68 mm, respectively, and the difference in ADD between the two periods of investigations was 152.57 DD. From March to May 2007, the mean BLs were 5.60, 7.50, 8.30, and 8.52 mm, respectively, and the sequential differences in ADDs were 110.22, 196.57, and 153.15 DD, respectively. Therefore, the larvae of the MG population can grow at a rate of approximately 0.75 mm per 100 DD (Table 3).

In the case of the MU population, the mean BLs of the larvae collected in September and October 2019 were 6.32 and 7.63 mm, respectively, and the difference in ADD was 235.65 DD. In April and May 2020, the mean BLs were 7.57 and 8.09 mm, respectively, and the difference in ADD was 32.39 DD. Upon investigating the newly hatched larvae after the adult flight period, the mean BLs of the larvae collected from August to October 2019 were 3.51, 5.39, and 7.09 mm, respectively. The differences in ADDs between the periods of investigations were 167.08 and 127.70 DD, respectively. The growth rate of the MU population larvae was 1.16 mm per 100 DD, which was 1.55 times faster than that of the MG population. The maximum BLs of the MG and MU populations were 10.69 and 9.29 mm, respectively, and when the growth rate above was applied, 1425.3 and 800.9 DD were required for the larvae at the MG and MU sites to grow from hatching to the maximum BL, respectively (Table 3).

The relationship between the flight patterns of *N. koreana* adults (pooled cumulative proportion of total flights) and DD was well described by the cumulative Weibull distribution model. The models explained 99.1% and 99.7% of the variations in flight patterns at MG and MU, respectively (MG: *r*^2^ = 0.991, *df* = 1, 9, *p* < 0.0001; MU: *r*^2^ = 0.994, *df* = 1, 10, *p* < 0.0001; Figure 8). The cumulative adult proportions of ADD at the two study sites were significantly different (two-sample Kolmogorov–Smirnov test, *ks* = 0.924, *p* < 0.0001; Figure 8). The ADDs required for the oviposition days predicted by the model were 789.11 and 207.68 DD, respectively. The oviposition dates corresponding to the ADD values above were 14 June 2005 at MG; and 20 June 2019 and 16 June 2020 at MU. Since 150 DDs is required from oviposition to hatching [35,38], the estimated hatching dates were 24 June 2006 at MG; and 7 July 2019 and 29 June 2020 at MU.

On average, the time at which the water temperature was lower than the T_b_ value was from December in MG, whereas in MU, it was from November; therefore, the period during which the larvae could grow was shorter in MU than in MG. The ADDs at the two study sites during the growth period from the estimated hatching day were 1882.5 and 1056.4 DD, respectively (Table 3).

## 4. Discussion

*N. koreana* larvae were divided into four groups by the Hcap program using HW (Figure 5), which is consistent with the results of a previous study [33]. The fact that the logarithms of HW fit a straight line against the number of groups indicates that the number of groups was well classified, without overlapping [46,48]. The linear regression equation derived in the present study is strong (*r*^2^ = 0.99, *p* < 0.01; Figure 5b). The excellent fit to the linear model indicated that there was no overlapping among larval groups. However, in the case of BL, the length increased in the MG population as the larvae developed. In group IV, the mean HW was larger in the MU population than in the MG population, whereas mean BL in the MG population was the largest (Figure 6, Table 2). In addition, since the ADDs from the estimated hatching date to winter of that year were 1882.5 DD and 1056.4 DD at MG and MU, respectively (Table 3), it was hypothesized that the BL of the MU population was lower because they have a semivoltine life cycle. When populations in very close proximity differ in voltinism, populations with different voltinism may show different body sizes [23]. For example, *Ischnura elegans* (Odonata: Coenagrionidae) and *Coenagrion puella* (Odonata: Coenagrionidae) are usually univoltine; however, some populations are semivoltine, showing different life histories. Although larval size was different between populations, the final-instar larval HW of the semivoltine population was higher than that of the univoltine population [53]. Similar results have been reported in *Calopteryx haemorrhoidalis* (Odonata: Calopterygidae) in Spain [54]. *C. puella* and *C. pulchellum* (Odonata: Coenagrionidae), damselfly species, reportedly exhibit faster growth and smaller adult sizes in the northern (Swedish) population than in the southern (Polish) population, which is also consistent with our findings [23]. These results demonstrate that even if the MU population is semivoltine, it does not adequately explain why the mean BL of the MU population was smaller than that of the MG population. *N. koreana* has a univoltine life cycle in South Korea [33]; therefore, we speculate that the MU population also has a univoltine life cycle, similar to the MG population, and has a compensating mechanism for cold water temperatures.

Our results suggest that *N. koreana* conforms to the converse Bergmann’s rule and does not follow the TSR. The BL of the population in cooler water temperatures was significantly shorter than that of the population in warmer water temperatures (Figure 6). Wild-caught adults of *Paropsis atomaria* (Coleoptera: Chrysomelidae) showed a clear trend of decreasing size with an increasing latitude. At the same time, they conformed to the TSR by demonstrating that larvae raised at higher temperatures resulted in smaller adults [13]. The authors concluded that the reason *P. atomaria* conforms to the converse Bergmann cline is because of local adaptation due to genetic differences between the populations, and *N. koreana* is considered to be a similar case. Notably, the HW of *N. koreana* larvae showed little change compared to BL, which showed a decrease at the colder water temperature. In insects, the instar number of larvae may vary depending on nutritional status; however, since the head capsule of an insect is a stable part that does not change, except for molting [55], BL is considered a more suitable criterion than HW for evaluating the body’s adaptation to temperature.

Recently, numerous studies have been conducted on voltinism, body size, sex, and trophic level as factors potentially influencing the direction and strength of temperature–size responses [17,18,56,57]. In particular, in the case of voltinism, larger species that are often univoltine with longer development times (ex. *Aquarius remigis* (Hemiptera: Gerridae)) tend to show converse Bergmann clines, whereas smaller species that are usually multivoltine with shorter development times (ex. *Drosophila melanogaster* (Diptera: Drosophilidae)) tend to show Bergmann clines [58]. In the Odonata taxon, a positive size–latitude relationship was reported for univoltine species, but it was negative for multivoltine species [59]. Such results support our finding that *N. koreana,* known to be a univoltine in Korea, conforms to the converse Bergmann’s rule.

In a previous study that investigated the growth rate of *N. koreana* larvae, a greenhouse imitating a small wetland was established in Boryeong-si, Chungcheongnam-do (36° 18 31.3′ N, 126° 37 26.1′ E), and larvae were introduced into it. The larval growth rate was estimated by randomly recapturing the larvae, measuring their size, and releasing them for 18 months. Because the study was conducted in an enclosed space without influx or disturbance of external *N. koreana* populations, the estimation of larval growth rate was considered accurate [35]. In the greenhouse, the annual mean water temperature was 17.9 °C and the larval growth rate was 0.70 mm per 100 DD. The results of our study in MG were similar to those of the greenhouse study in Boryeong-si, with an average annual water temperature of 21.07 °C and a larval growth rate of 0.75 mm per 100 DD. However, the population at MU with an annual average water temperature of 12.60 °C had a higher larval growth rate of 1.16 mm per 100 DD. In particular, season length is a key factor influencing the direction of Bergmann’s rule in arthropods [6,10]. The T_b_ value of *N. koreana* was 14.3 °C [33,38], and the period when the water temperature was above the T_b_ at MU was from mid-April to the end of September, whereas at MG, the larvae would probably continue to feed and develop from the beginning of March to the end of October. Considering that the larger larval body size prior to winter probably results in higher survival rates during winter diapause, rapid larval growth is clearly beneficial for larval development at lower water temperatures [56]. Therefore, it is presumed that the MU population compensates for the seasonal limitation of low water temperature by growing faster than the MG population. *Lestes viridis* (Odonata: Lestidae) accelerated their growth rate to emerge in time, when experiencing the end of growth season [60]. A similar species, *L. sponsa* (Odonata: Lestidae), a strictly univoltine damselfly in Europe, emerged at smaller sizes when high-latitude larvae grew faster than low-latitude population, but lacked compensation for the shorter season [61,62]. This is consistent with our findings, suggesting that temperature and season length considerably influence the growth and development of larvae in the field [62]. The compensation for the lower temperature at higher latitudes denotes a decrease in voltinism [50]. Decreasing season length leads to a decrease in body size up to the point where another year is added to the length of the life cycle prior to being an adult; at that point, the body size increases. This pattern is referred to as U-shaped or sawtooth cline, and has been observed in several insect species with variable voltinism [7,62]. Our findings suggest that the MU population maintains a univoltine life cycle and is in a stage prior to voltinism decline (shift to semivoltine).

According to the estimated larval growth rate, *N. koreana* larvae seem to be able to grow to their maximum size after hatching before winter, but this is not the case. Freshwater ecosystems (e.g., lakes, ponds, and wetlands) can be more isolated and fragmented than terrestrial and marine habitats [63], and the community structure of lotic ecosystems can be strongly influenced by physical disturbances, such as droughts [64]. In addition to temperature, various factors, such as dissolved oxygen, food quality, and predation pressure, affect the growth of aquatic insects [65,66,67]. *N. koreana* mainly inhabits wetlands where the water depth is less than 10 cm [31], and many factors can inhibit its growth. Therefore, it cannot be presumed that larvae will become adults before winter, because the ADD from hatching until winter is greater than the ADD required for larvae to reach their maximum BL after hatching. If there is a risk of drought before emergence, the larvae can grow in winter [50]. However, in Korea, winter is the season with the least amount of precipitation and precipitation increases in summer, the period of adult activity. Also, *N. koreana* is a species adapted to a high temperature range, with a T_b_ of 14.3 °C, as the genus *Nannophya* is widely distributed in tropical and subtropical areas [31,33,38]. Therefore, in the Korean climate, *N. koreana* larvae will not grow in winter, and it is estimated that it will complete the univoltine life cycle by hatching in summer and growing over the following spring.

Most studies on Bergmann’s cline or TSR have analyzed variations in the body size of organisms in relation to latitude or altitude, along with various factors [7,15,24,57,58,68]. Although body size is also affected by latitude-dependent photoperiod, as well as temperature [16], our investigations were conducted in a geographically small zone, with few possibilities of latitude or altitude influencing the size. In addition, although the location and investigation periods of the two study sites were different, temperature is the most important factor influencing the growth of insects [64], and in both areas, there was no notable environmental difference other than the wetland sources, which is the cause of the difference in water temperature. Therefore, we basically performed the analysis with a focus on the relationship between the field water temperature and the larval body size. Even if the distance between habitats is geographically far enough to cause differences in temperature or photoperiod, there can be variations in body size, suggesting that it is essential to understand and consider the unique environmental characteristics of habitats when examining variations in organisms between different geographical zones [69].

## 5. Conclusions

We compared larval body sizes of *N. koreana* populations in two regions with different water temperatures. There was little difference in HW between the two regions; however, BL was lower in wetlands with lower water temperatures. Our results showed that *N. koreana* has the opposite result from Bergmann’s rule and TSR, a widely known ecological perspective that explains smaller body sizes of organisms growing at high temperatures. The converse Bergmann’s rule is more commonly reported in univoltine species than in multivoltine species, which may be one of the reasons for the converse Bergmann’s rule in *N. koreana*, which is known to be univoltine in Korea. The larval growth rate was higher in wetlands with lower water temperatures than in wetlands with higher water temperatures, which is speculated to be a mechanism for compensating for the shorter growth period in wetlands with lower water temperatures. Many studies have used Bergmann’s rule or TSR to explain the relationship between geographical distribution by latitude and body size in organisms; however, our study established the relationship between temperature and body size in two wetlands with a clear difference in water temperature despite being geographically close. The results suggest that it is important to examine detailed factors, such as habitat type and environment, as well as simple geographical location when conducting body size comparisons based on temperature. The distinct temperature–body size relationship in *N. koreana* is attributable to local adaptation. Since Korea is the northern limit of *N. koreana* distribution, our findings provide unique information on various ecological responses that can be observed within the distribution range.

## Figures and Tables

**Figure 1 biology-11-00830-f001:**
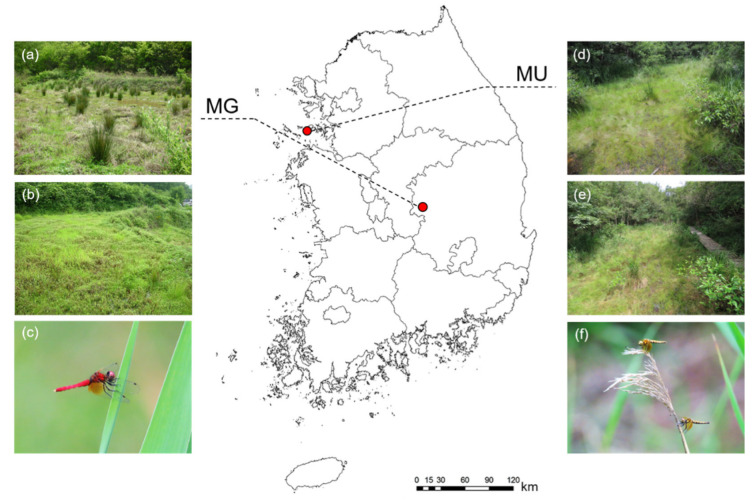
Two study sites selected for *Nannophya koreana* investigation in Korea. The first site was located in Mungyong-si (MG) (36° 33 15.9′ N, 128° 00 20.2′ E) (**a**,**b**) and the other site was located in Muui-do (MU) (37° 22 57.2′ N, 126° 24 53.4′ E) (**d**,**e**). Pictures of male (**c**) and female (**f**) adults of *N. koreana*.

**Figure 2 biology-11-00830-f002:**
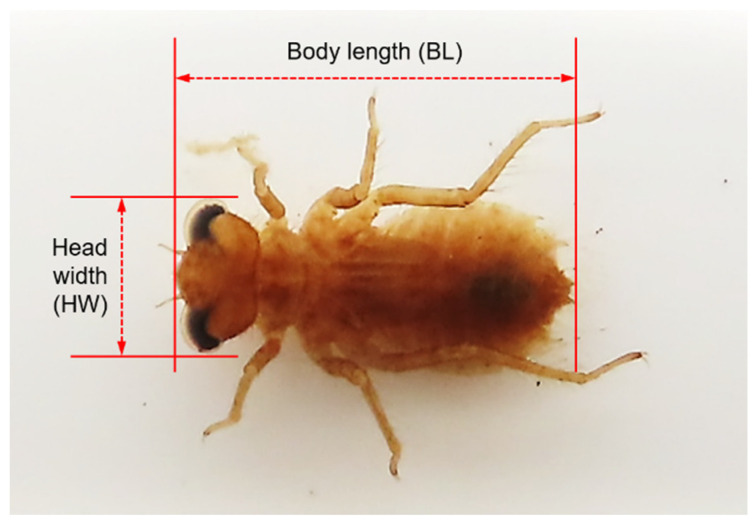
The body size measurement standards of *Nannophya koreana*; head width (HW) and body length (BL).

**Figure 3 biology-11-00830-f003:**
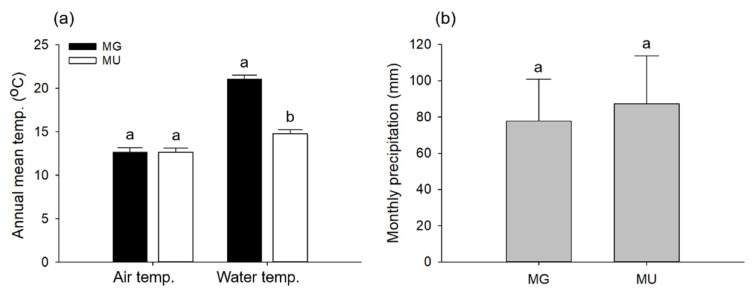
Comparison of annual mean temperature (**a**) and monthly precipitation (**b**) in the two study areas during the overlapping study period (MG: from August 2006 to July 2007, and MU: from August 2019 to July 2020). Bars represent standard error of the mean. Different letters indicate statistically significant differences (*t*-test, *p* < 0.05).

**Figure 4 biology-11-00830-f004:**
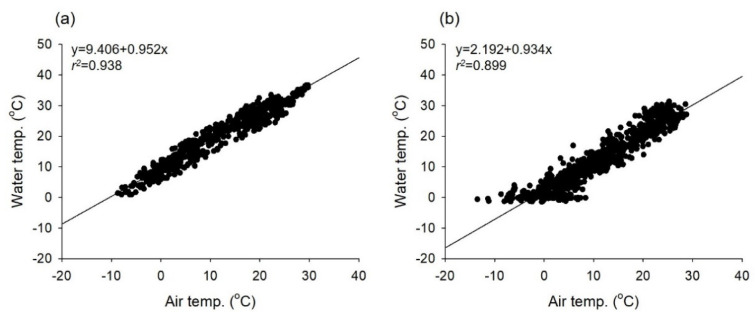
Correlation between air temperatures from weather stations and water temperatures recorded using data loggers: (**a**) MG; June 2006−2007, and (**b**) MU; 2019−2021.

**Figure 5 biology-11-00830-f005:**
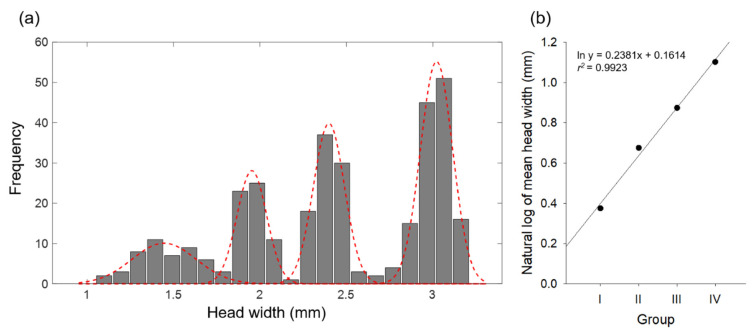
The frequency distribution of observed head width of *Nannophya koreana* larvae (**a**). The dotted lines represent the larval group distributions. Regression relationship between the natural logarithm of the mean larval head widths and the number of groups (**b**).

**Figure 6 biology-11-00830-f006:**
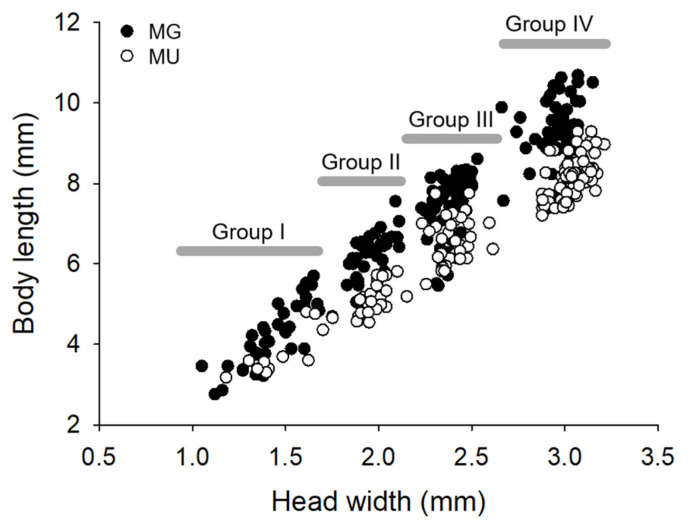
The relationship between head width and body length of *Nannophya koreana* larvae at the two study sites. The larvae were divided into four groups according to their size.

**Figure 7 biology-11-00830-f007:**
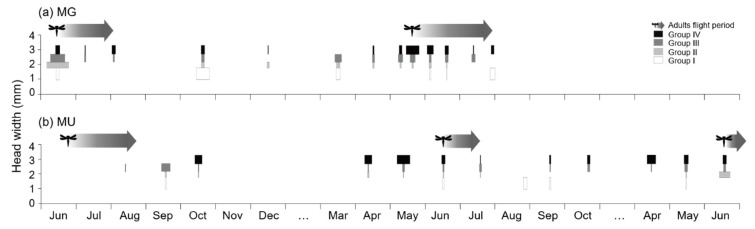
The size–frequency distribution of *Nannophya koreana* larvae based on the groups using Hcap analysis. Arrows present the flight period of *N. koreana* adults. Width of horizontal white, light-gray, dark-gray, and black color bars indicate the proportions of larvae in Group I, II, III, and IV, respectively. The study period was from August 2006 to July 2007 in MG (**a**) and from June 2019 to June 2021 in MU (**b**).

**Figure 8 biology-11-00830-f008:**
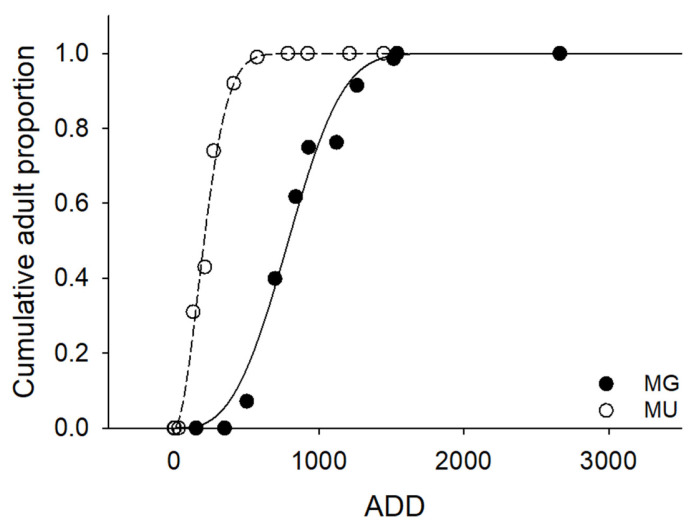
The cumulative proportion of *Nannophya koreana* adults at the two study sites (closed circle: MG, Mungyong-si; open circle: MU, Muui-do) during the study periods. The solid and dashed lines indicate the flight proportions predicted against ADD using 14.3 °C as the baseline temperature: the equations of the solid and dashed lines are $f\left(x\right)=1-\exp\left[-{\left(x⁄889.05\right)}^{3.07}\right]$ and $f\left(x\right)=1-\exp\left[-{\left(x⁄252.31\right)}^{1.88}\right]$, respectively.

**Table 1 biology-11-00830-t001:** Mean head widths, ranges, and misclassification probabilities of the four larval groups of *Nannophya koreana* calculated using the Hcap program and Dyar’s ratio.

Group	*n*	Head Width (mm)		Probability of Misclassification			Dyar’s Ratio
		Mean ± SE	Range	*i* as *i* − 1	*i* as *i* + 1	Total	
I	48	1.45 ± 0.17	1.05−1.75	0.0000	0.0433	0.0433	-
II	61	1.97 ± 0.08	1.83−2.15	0.0129	0.0068	0.0197	1.35
III	90	2.40 ± 0.09	2.23−2.67	0.0046	0.0004	0.0050	1.22
IV	131	3.01 ± 0.09	2.74−3.21	0.0003	0.0000	0.0003	1.26

**Table 2 biology-11-00830-t002:** Comparison of head width (HW) and body length (BL) of *Nannophya koreana* larvae at the two study sites (mean ± SE).

Standard	Site	Group I	Group II	Group III	Group IV
HW	MG	1.44 ± 0.17	1.95 ± 0.08	2.40 ± 0.09	2.97 ± 0.08 *
	MU	1.49 ± 0.18	1.98 ± 0.06	2.40 ± 0.08	3.04 ± 0.08
BL	MG	4.28 ± 0.78	6.18 ± 0.60 *	7.62 ± 0.75 *	9.26 ± 0.74 *
	MU	3.86 ± 0.58	5.17 ± 0.37	6.57 ± 0.55	8.21 ± 0.48

* indicates statistically significant differences (*t*-test, *p* < 0.01).

**Table 3 biology-11-00830-t003:** The growth rate and maximum body length (BL) of *Nannophya koreana* larvae and the accumulated degree day (ADD) requirement from hatching to maximum size, and the ADD from hatching to winter at the two study sites (MG: Mungyong-si; MU: Muui-do).

Site	Growth Rate (mm/100 DD)	Maximum BL (mm)	Required ADD from Hatching to Maximum BL	ADD from Hatching until Winter
MG	0.75	10.69	1425.3	1882.5
MU	1.16	9.29	800.9	1056.4

## Data Availability

Not applicable.

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
