# Peer review of "Ecological Responses of Nannophya koreana (Odonata: Libellulidae) to Temperature: Following Converse Bergmann’s Rule"

_biology, 2022, doi:10.3390/biology11060830_

Round 1
Reviewer 1 Report
These are my main comments on the manuscript (Biology-1726535) entitled “Ecological responses of Nannophya koreana (Odonata: Libellulidae) to temperature: following converse Bergmann’s rule”. The manuscript investigates the biological responses of N. koreana by comparing the body size in two habitats. The results showed the relationship between temperature and body size of N. koreana in two wetlands with difference in water temperatures reflecting details as habitat types and the surrounding environment. Following moderated revisions should be incorporated in the manuscript prior to acceptance.
L.22: Delete “In this article,”
Ls.25-28: Summarize this sentence
L.35: …degree-days…
L.21: Delete “significantly”
Ls.44-45: Keywords should be in alphabetic order. Also, keywords serve to widen the opportunity to be retrieved from a database. To put words that already are into title and abstracts makes KW not useful. Please choose terms that are neither in the title nor in abstract.
Ls.91-99: In addition, a hypothesis for this study is needed.
Ls.106, 107 and 100: Delete “approximately”
Ls.200-208: For each statistical analysis, provide the t-value and degree freedom.
Ls.213-214: This information should be in material and methods section.
Ls.241-251: Again, for each statistical analysis, provide the t-value and degree freedom.
L.246: Change “were” by “was”.
L.276: The size-frequency distribution…
Ls.293-294: Which differences? Rephrase
L.295: Again, delete “approximately”
Ls.338-340: Rephrase this sentence
L.395: …had a higher larval…
Author Response
These are my main comments on the manuscript (Biology-1726535) entitled “Ecological responses of Nannophya koreana (Odonata: Libellulidae) to temperature: following converse Bergmann’s rule”. The manuscript investigates the biological responses of N. koreana by comparing the body size in two habitats. The results showed the relationship between temperature and body size of N. koreana in two wetlands with difference in water temperatures reflecting details as habitat types and the surrounding environment. Following moderated revisions should be incorporated in the manuscript prior to acceptance.
Answer:
- Thank you for your comments. We have made changes to reflect the suggestions provided by the reviewer. We have marked the revisions with 'track changes' in the manuscript. We have provided point-by-point responses to the reviewer’s comments and concerns. All line numbers refer to the numbers in the revised manuscript file with tracked changes.
L.22: Delete “In this article,”
Answer:
- Thank you for your comments. This has been corrected in the revised manuscript (line 22).
Ls.25-28: Summarize this sentence
Answer:
- Thank you for this suggestion. This has been summarized in the manuscript (lines 24-27).
L.35: …degree-days…
Answer:
- Thank you for your comments. This has been corrected in the revised manuscript (Line 32).
L.21: Delete “significantly”
Answer:
- Thank you for your comments. This has been corrected in the revised manuscript (line 18).
Ls.44-45: Keywords should be in alphabetic order. Also, keywords serve to widen the opportunity to be retrieved from a database. To put words that already are into title and abstracts makes KW not useful. Please choose terms that are neither in the title nor in abstract.
Answer:
- Thank you for this suggestion. This has been corrected in the revised manuscript (line 41).
Ls.91-99: In addition, a hypothesis for this study is needed.
Answer:
- Thank you for your comments. This has been corrected in the revised manuscript (line 107-109).
Ls.106, 107 and 100: Delete “approximately
Answer:
- Thank you for your comments. This has been corrected in the revised manuscript (lines 122, 123, 125 and 127).
Ls.200-208: For each statistical analysis, provide the t-value and degree freedom.
Answer:
- Thank you for your comments. This has been corrected in the revised manuscript (lines 229, 232, 234 and 235).
Ls.213-214: This information should be in material and methods section.
Answer:
- Thank you for your comments. As this information was already included in the Materials and Methods section (lines 152-156), we deleted this sentence from the Results.
Ls.241-251: Again, for each statistical analysis, provide the t-value and degree freedom.
Answer:
- Thank you for your comments. This has been corrected in the revised manuscript (lines 271-280).
L.246: Change “were” by “was”.
Answer:
- We corrected “were” by “was” according to your comment, but after proofreading in English, the verb was again changed to “were” (line 272).
L.276: The size-frequency distribution…
Answer:
- Thank you for your comments. This has been corrected in the revised manuscript (Line 305).
Ls.293-294: Which differences? Rephrase
Answer:
- Thank you for your comments. This has been corrected in the manuscript (lines 323-324).
L.295: Again, delete “approximately”
Answer:
- Thank you for your comments. This has been corrected in the manuscript (line 325).
Ls.338-340: Rephrase this sentence
Answer:
- Thank you for your comments. This has been corrected in the revised manuscript (lines 358-340).
L.395: …had a higher larval…
Answer:
- Thank you for your comments. This has been corrected in the revised manuscript (line 418).
Reviewer 2 Report
I have just finished reviewing the manuscript “Ecological responses of (Odonata: Libellulidae) to temperature: following converse Bergmann’s rule”. The authors studied thermal effects on protected and endangered odonate Nannophya koreana. They measured larval growth and development in natural conditions. The results confirm a commonly reported in ectotherms converse Bergmann’s rule and no support for temperature-size-rule. My general impression about the manuscript is positive, although the findings are not really novel. However, the text needs to be revisited in order to increase its quality. Adding more details about odonate-specific life cycle regulation would deepen the insights of the species responses to environmental constrains, such as suboptimal temperatures. Below more detailed comments and suggestions how to improve the text.
ABSTRACT
I suggest to start this section with the second sentence “Ecological rules such as Bergmann's rule and...” and then species description.
The abstract could by shortened, e.g. by removing details about technical part (L.30), “…using an image analysis program.”.
INTRODUCITON
Say whether larger or smaller body size is advantageous for insects in general, and odonates in particular. Now nothing about it in the Introduction.
L55-56 I agree, but in organisms with variable voltinism, e.g. odonates, these rules might not work. Please refer to this in the Introduction.
L61 A very relevant recent paper about a univoltine odonate with a large geographic/latitudinal distribution that I suggest to cite here is by Johansson et al. 2021 https://doi.org/10.1111/evo.14147 The authors show both thermal effects as well as quantitative genetic background behind larval size. The results somewhat a agree with the current results, confirming converse Bergmann’s cline.
L79 “…different ecological pattern from previous research results.” – add reference(s).
L93-98 Testing responses to low temperature comes out of blue. Especially that in the first part of the Introduction the authors focus on global warming. Better to extend the Introduction by adding a clear hypothesis linked to habitat-specific temperature and predicted N. koreana responses in terms of body size.
In insects, developmental compensation might be expressed by adding another growth season for completing one generation. This happens unless a species is obligatory univoltine (i.e., voltinism is fixed). I suggest to mention about voltinism in the Introduction sections. Otherwise unclear what kind of compensation the authors mention about.
MATERIALS AND METHODS
L100 I really recommend to at least mention about seasonal regulation of larval development in odonates. A recently published review on this topic is by Norling 2021 (https://files.wachholtz-verlag.de/openaccess/ijo/24/10-23797-2159-6719_24_1.pdf). The author discusses the importance of temperature, photoperiod (the later omitted in the manuscript) and interaction of the two in shaping larval development and growth in temperate odonates, which is very relevant to the current study.
Note that each type of sampling sites was not replicated. Considering that each site is different and unique, there might be other environmental factors that could affect larval size. But these other factors were not controlled.
L110 you mean “northwest”?
L142 Because each site was sampled at different dates, this could also affect the results. Such season-specific environmental conditions, and how these season-to-season environmental changes affect odonate final size, were shown and discussed in Sniegula et al. 2016.
I recommend to add this information and reference in the Discussion.
L153 Please explain why you chose these traits. Do these traits translate into overall structural body size? How did the authors measure body length, did it include cerci/anal appendages? Adding a photo with larval measurement would improve method description.
L171 Why not analysing growth rate based on head width – a commonly measured trait in odonate life history studies?
L210 Describe what error bars are presented in Fig. 2.
L328-335 Sorry, but in my opinion, this is a repetition from methods. I suggest the whole paragraph.
L349 Variable voltinism might explain these size differences. Voltinism should be described in the Introduction, and more so in Methods (see previous comments).
L351-353 A very relevant recent paper by Sniegula et al. 2012 (https://onlinelibrary.wiley.com/doi/full/10.1111/j.1600-0706.2011.20015.x) showing this pattern based on both field and laboratory data. I suggest to cite it here.
L359-360 Well, ideally this should be checked under controlled laboratory conditions. Otherwise speculative.
L371-372 I suggest to add that this explanation is likely less relevant to insect groups with variable instar number, like in odonates. Here cited Nijhout 1975 studied Manduca sexta with fixed instar number.
L357 You mean “temperature response”?
L390-392 Unclear. Please rephrase.
L399-403 Again, strong empirical support for the compensation plus in-depth discussion in Sniegula et al. 2016 (http://onlinelibrary.wiley.com/doi/10.1111/een.12314/abstract).
L413 Here a discussion of larval winter diapause development should be added. The classic and most recent original paper on the topic can be found in Norling 2021 (https://files.wachholtz-verlag.de/openaccess/ijo/24/10-23797-2159-6719_24_1.pdf).
L418-425 As mentioned, the here presented results for each of the two study sites could be strongly affected by between-season variation in thermal conditions (i.e., different sampling dates, and hence temperatures, for each site). Discuss it.
L432-434 The authors never mention about semivoltine and partivoltine life cycles, which are common in odonates from temperate regions. I recommend to broaden the discussion by adding these.
Author Response
I have just finished reviewing the manuscript “Ecological responses of (Odonata: Libellulidae) to temperature: following converse Bergmann’s rule”. The authors studied thermal effects on protected and endangered odonate Nannophya koreana. They measured larval growth and development in natural conditions. The results confirm a commonly reported in ectotherms converse Bergmann’s rule and no support for temperature-size-rule. My general impression about the manuscript is positive, although the findings are not really novel. However, the text needs to be revisited in order to increase its quality. Adding more details about odonate-specific life cycle regulation would deepen the insights of the species responses to environmental constrains, such as suboptimal temperatures. Below more detailed comments and suggestions how to improve the text.
Answer:
- We are grateful to the reviewers for their insightful comments on the manuscript. We have made changes to reflect the suggestions provided by the reviewer. We have marked the revisions with 'track changes' in the manuscript. Given below are the point-by-point responses to the reviewer’s comments and concerns. All line numbers refer to the numbers in the revised manuscript file with tracked changes.
ABSTRACT
I suggest to start this section with the second sentence “Ecological rules such as Bergmann's rule and...” and then species description.
Answer:
- Thank you for this suggestion. This has been corrected in the revised manuscript (lines 22-27).
The abstract could by shortened, e.g. by removing details about technical part (L.30), “…using an image analysis program.”.
Answer:
- Thank you for your comments. This has been corrected in the revised manuscript (Line 28).
INTRODUCITON
Say whether larger or smaller body size is advantageous for insects in general, and odonates in particular. Now nothing about it in the Introduction.
Answer:
- Thank you for this suggestion. We have added a more detailed explanation to the manuscript (lines 70-75).
L55-56 I agree, but in organisms with variable voltinism, e.g. odonates, these rules might not work. Please refer to this in the Introduction.
Answer:
- Thank you for this suggestion. We have referred to the variability of voltinism in the manuscript (lines 53-56).
L61 A very relevant recent paper about a univoltine odonate with a large geographic/latitudinal distribution that I suggest to cite here is by
Johansson et al. 2021 https://doi.org/10.1111/evo.14147
The authors show both thermal effects as well as quantitative genetic background behind larval size. The results somewhat a agree with the current results, confirming converse Bergmann’s cline.
Answer:
- Thank you for your comments. We have added a reference paper to the manuscript (line 60).
L79 “…different ecological pattern from previous research results.” – add reference(s).
Answer:
- We added the reference paper in the manuscript (line 90).
L93-98 Testing responses to low temperature comes out of blue. Especially that in the first part of the Introduction the authors focus on global warming. Better to extend the Introduction by adding a clear hypothesis linked to habitat-specific temperature and predicted N. koreana responses in terms of body size.
Answer:
- Thank you for your comments. This has been corrected in the revised manuscript (lines 107-109).
In insects, developmental compensation might be expressed by adding another growth season for completing one generation. This happens unless a species is obligatory univoltine (i.e., voltinism is fixed). I suggest to mention about voltinism in the Introduction sections. Otherwise unclear what kind of compensation the authors mention about.
Answer:
- Thank you for this suggestion. We have mentioned voltinism in the manuscript (lines 53-56, 63-69).
MATERIALS AND METHODS
L100 I really recommend to at least mention about seasonal regulation of larval development in odonates. A recently published review on this topic is by Norling 2021 (https://files.wachholtz-verlag.de/openaccess/ijo/24/10-23797-2159-6719_24_1.pdf). The author discusses the importance of temperature, photoperiod (the later omitted in the manuscript) and interaction of the two in shaping larval development and growth in temperate odonates, which is very relevant to the current study.
Answer:
- Thank you for your suggestion. We have added a reference paper in the manuscript, supporting the fact that larval development is affected by temperature and photoperiod (lines 190-195).
Note that each type of sampling sites was not replicated. Considering that each site is different and unique, there might be other environmental factors that could affect larval size. But these other factors were not controlled.
Answer:
- Thank you for your comments. Temperature was the most critical factor affecting insect development, and the two regions had no significant environmental differences other than wetland sources, which led to differences in water temperature. Therefore, our analysis was based on the relationship between water temperature of the field(site) and larval body size. We have added relevant explanations to the Discussion (lines 463-468).
L110 you mean “northwest”?
Answer:
- We apologize for this confusion. This has been corrected in the revised manuscript (line 125).
L142 Because each site was sampled at different dates, this could also affect the results. Such season-specific environmental conditions, and how these season-to-season environmental changes affect odonate final size, were shown and discussed in Sniegula et al. 2016.
I recommend to add this information and reference in the Discussion.
Answer:
- Thank you for your comments. This has been corrected in the Discussion (lines 428-440).
L153 Please explain why you chose these traits. Do these traits translate into overall structural body size? How did the authors measure body length, did it include cerci/anal appendages? Adding a photo with larval measurement would improve method description.
Answer:
- koreana is an endangered species that is legally protected in Korea. During on-site investigations, photographs of the larvae were captured while the larvae were alive and released as soon as possible to protect the population. In this way, there may be errors in measuring the size of small body parts; therefore, we selected head width and body length as the criteria that are considered to have the least error during measurement. In addition, in a previous study on N.koreana, the life history and growth rate were estimated using head width and body length. In this study, the same standards were applied to compare the results of the previous study (lines 197-198), and the body size measurement method and photos were added to the manuscript (lines 168-171, 172-175).
L171 Why not analysing growth rate based on head width – a commonly measured trait in odonate life history studies?
Answer:
- In a previous study, body length was used to classify larval cohorts according to Cassie’s (1954) method, and was also applied as the standard for growth rate analysis. In this study, the same standards were applied to compare the results with those of a previous study. Interestingly, when the results of the two regional populations were compared, the range of head width was similar, but that of body length was different. Relevant information has been included in the text (line 392-397).
L210 Describe what error bars are presented in Fig. 2.
Answer:
- Thank you for your comments. This has been corrected in the revised manuscript (line 239).
L328-335 Sorry, but in my opinion, this is a repetition from methods. I suggest the whole paragraph.
Answer:
- Thank you for this suggestion. We agree with your opinion and have deleted this paragraph from the manuscript.
L349 Variable voltinism might explain these size differences. Voltinism should be described in the Introduction, and more so in Methods (see previous comments).
Answer:
- Thank you for your comments. We have added content and references related to the Introduction and Materials and Methods in response to your comments (lines 53-56, 63-69, 190-195).
L351-353 A very relevant recent paper by Sniegula et al. 2012 (https://onlinelibrary.wiley.com/doi/full/10.1111/j.1600-0706.2011.20015.x) showing this pattern based on both field and laboratory data. I suggest to cite it here.
Answer:
- Thank you for this suggestion. We have added the contents and references to the manuscript (lines 375-378).
L359-360 Well, ideally this should be checked under controlled laboratory conditions. Otherwise speculative.
Answer:
- Thank you for your comments. We have corrected this verb in the sentence (line 381).
L371-372 I suggest to add that this explanation is likely less relevant to insect groups with variable instar number, like in odonates. Here cited Nijhout 1975 studied Manduca sexta with fixed instar number.
Answer:
- Thank you for your comments. We have added the contents and references to the manuscript (lines 394-395).
L357 You mean “temperature response”?
Answer:
- We apologize for this confusion. We have confirmed that there was an error in the sentence and corrected it in the manuscript (lines 399-400).
L390-392 Unclear. Please rephrase.
Answer:
- It has been corrected in the manuscript (lines 415-417).
L399-403 Again, strong empirical support for the compensation plus in-depth discussion in Sniegula et al. 2016 (http://onlinelibrary.wiley.com/doi/10.1111/een.12314/abstract).
Answer:
- Thank you for your comments. We have added the contents and references to the manuscript (lines 428-440).
L413 Here a discussion of larval winter diapause development should be added. The classic and most recent original paper on the topic can be found in Norling 2021 (https://files.wachholtz-verlag.de/openaccess/ijo/24/10-23797-2159-6719_24_1.pdf).
Answer:
- Thank you for comment. We have added content about the larval growth potential during winter and the references in the manuscript (lines 451-458).
L418-425 As mentioned, the here presented results for each of the two study sites could be strongly affected by between-season variation in thermal conditions (i.e., different sampling dates, and hence temperatures, for each site). Discuss it.
Answer:
- Thank you for your comments. We have added the interpretations and explanations for the different sampling periods at the two study sites (lines 463-472).
L432-434 The authors never mention about semivoltine and partivoltine life cycles, which are common in odonates from temperate regions. I recommend to broaden the discussion by adding these.
Answer:
- Thank you for your comments. We have added an interpretation of the possibility of semivoltine of koreana in the manuscript while adding content about variable voltinism of odonates (lines 434-440).
Reviewer 3 Report
Cha Young Lee and colleagues present a nice study on temperture size development in a species of dragonflies. They found conversed Bergmans rule for the larvae of this species. I like the overall study and, besides minor aspects, have only few objections (primarily the argumentation of growth, development duration and imaginal flight and their interconnections nedd to be more on point), but have my doubts whether this manuscript is actually suitable for the journal Biology. It is rather specialised and focuses on a narrow subject with only minor importance for overall biological phenomena. Therefore I suggest to reject the paper and the authors to submit it to a more specialised journal (e.g. Physiological Entomology, MDPI Insects, etc)
Minor comments for optimisation in a potential new submission:
-keywords: should not double with the title as this does not increase visibility; use unique keywords
-literature: some publications of matter are missing, e.g.
Shelomi, M. (2012). Where are we now? Bergmann's rule sensu lato in insects. Am. Nat. 180, 511–519. doi: 10.1086/667595
https://www.frontiersin.org/articles/10.3389/fevo.2017.00025/full#B43
Small disclaimer: the software says "Reject means tha article has serious flaws, etc.: that is not the case, the only major flaw is, that it does not (in my oppinion) meet the criteria for this particular journal.
Author Response
Cha Young Lee and colleagues present a nice study on temperature size development in a species of dragonflies. They found conversed Bergmann’s rule for the larvae of this species. I like the overall study and, besides minor aspects, have only few objections (primarily the argumentation of growth, development duration and imaginal flight and their interconnections need to be more on point), but have my doubts whether this manuscript is actually suitable for the journal Biology. It is rather specialised and focuses on a narrow subject with only minor importance for overall biological phenomena. Therefore I suggest to reject the paper and the authors to submit it to a more specialised journal (e.g. Physiological Entomology, MDPI Insects, etc)
Small disclaimer: the software says "Reject means the article has serious flaws, etc.: that is not the case, the only major flaw is, that it does not (in my opinion) meet the criteria for this particular journal.
Answer:
- Nannophya koreana has a limited distribution in Korea and Japan; therefore, there is a high risk of a decreasing population. However, as it has only been two years since koreana was separated from N. pygmaea, the extinction risk of N. koreana has not yet been evaluated by the IUCN. To establish standards for species conservation, various ecological data, such as the results of this study, are needed. Previous studies related to converse Bergman's rule were conducted in relation to latitude on a geographically broad scale, our study described converse Bergman's rule, which can also occur in a local area. We supplemented our manuscript with the interpretation of N. koreana voltinism and compensation for the length of the growing season. Our findings are a unique study of the ecological response of the dragonfly to temperature, and at the same time, can provide important information for the conservation of endangered species, so it is suitable for the Conservation section of Biology.
Minor comments for optimisation in a potential new submission:
-keywords: should not double with the title as this does not increase visibility; use unique keywords
Answer:
- Thank you for this suggestion. This has been corrected in the revised manuscript (line 41).
-literature: some publications of matter are missing, e.g.
Shelomi, M. (2012). Where are we now? Bergmann's rule sensu lato in insects. Am. Nat. 180, 511–519. doi: 10.1086/667595
https://www.frontiersin.org/articles/10.3389/fevo.2017.00025/full#B43
Answer:
- Thank you for your delicate comment. We added the reference in the manuscript (line 61).
Round 2
Reviewer 2 Report
In general I am happy with the revised version of the manuscript.
After English editing of the revised sections, the text should be further processed towards publishing.
Reviewer 3 Report
The minor comments were amended. I still think the manuscript is not really suitable for the scope of Biology.
I did argue this in the previous round of reviews and two other reviewers as well as the editor seem to disagree. Therefore, I have no objections against accepting the manuscript for publication, besides the fact, that I think it would be more suitable in Insects or a similar journal with a narrower scope.